# THC and CBD: Villain versus Hero? Insights into Adolescent Exposure

**DOI:** 10.3390/ijms24065251

**Published:** 2023-03-09

**Authors:** Nicholas Pintori, Francesca Caria, Maria Antonietta De Luca, Cristina Miliano

**Affiliations:** 1Department of Biomedical Sciences, University of Cagliari, 09042 Cagliari, Italy; 2School of Neuroscience, Virginia Polytechnic Institute and State University, Blacksburg, VA 24061, USA

**Keywords:** THC, CBD, adolescence

## Abstract

Cannabis is the most used drug of abuse worldwide. It is well established that the most abundant phytocannabinoids in this plant are Δ9-tetrahydrocannabinol (THC) and cannabidiol (CBD). These two compounds have remarkably similar chemical structures yet vastly different effects in the brain. By binding to the same receptors, THC is psychoactive, while CBD has anxiolytic and antipsychotic properties. Lately, a variety of hemp-based products, including CBD and THC, have become widely available in the food and health industry, and medical and recreational use of cannabis has been legalized in many states/countries. As a result, people, including youths, are consuming CBD because it is considered “safe”. An extensive literature exists evaluating the harmful effects of THC in both adults and adolescents, but little is known about the long-term effects of CBD exposure, especially in adolescence. The aim of this review is to collect preclinical and clinical evidence about the effects of cannabidiol.

## 1. Introduction

### 1.1. History and Discovery of Cannabis

Cannabis is a plant belonging to the *Cannabaceae* family, which originally grew in Central Asia. From 4000 B.C., Asian populations consumed its fruit and used its fibers to make textiles, ropes, and paper [1,2]. The man who discovered the therapeutic potential of this weed domesticated it to use it as a medicinal herbal drug to treat symptoms such as nausea, migraine, intestinal constipation, and rheumatic pain [3,4,5]. The first evidence of cannabis being used as a recreational drug was reported in 400 B.C. by the Greek historian Herodotus, but it was only in 800 A.D. that smoking cannabis became more common in the Middle East and South Asia [5]. At that time, alcohol, tobacco, and coffee were predominant in Europe, while cannabis had been used as a staple in industry and manufacturing in Europe and North and South America since the 1500s and as a medicine throughout the 1800s worldwide. It was only in the 20th century that cannabis recreational use arose, and many countries scheduled it as an illicit drug [1,2].

Cannabis’ composition has been extensively studied in the last century, and we know it contains more than 120 phytocannabinoids [6,7,8]. Among them, cannabidiol (CBD) [6] and Δ9-tetrahydrocannabinol (THC) [9] are the most abundant compounds and therefore the best known. These two compounds are remarkably similar in their chemical structures (see Figure 1); indeed, they bind to the same cannabinoid receptors (CB1 and CB2) while leading to completely different effects (for details, see Section 1.3).

An extensive literature has demonstrated that chronic cannabis use is harmful for the pulmonary and respiratory system [10,11], the cardiovascular system [12,13], as well as the central nervous system (CNS) [14,15,16,17,18], especially if used in adolescence [19,20,21,22,23]. Moreover, the efforts made throughout the years by companies and criminal organizations to make cannabis more potent have led to very high THC levels in weed strains, deeply influencing consumption patterns and long-term use outcomes [24]. Nevertheless, starting from the early 2000s, many countries have legalized cannabis for medical purposes and some of them even for recreational use. As a result of this, in some parts of the world you can now order marijuana via the Internet or using a smartphone app [25]. Currently, in the US (as January 2023) 45 out of 51 states have legalized medical marijuana, while 22 of these states have approved it for recreational use as well (https://disa.com/maps/marijuana-legality-by-state; accessed on 31 January 2023). While cannabis has been fully legal in Canada since 2018, all the other continents (South America, Europe, and Australia) have legalized it only for medical purposes, with some exceptions. Moreover, a variety of hemp-based products, including CBD and THC, have been widely available in the food and health industry since 2018, when the marketing of such products was legalized by the Agricultural Improvement Act [26]. The legalization of cannabis is mainly due to political and social factors pushing for the decriminalization of marijuana possession to decrease prison overcrowding and to avoid the issuance of criminal records. The legalization of medical and recreational use plus the wide availability of products has clearly shifted the perceptions of cannabis’ effects among people, especially youths, albeit there are no data confirming an increase in use due to legalization [27,28,29,30,31].

### 1.2. Epidemiology: Use of Cannabis in Adolescence

In 2020, cannabis was the most used drug globally, with 209 million users, and the percentage of use in adolescence grows every year [32]. Figure 2 shows the global prevalence of cannabis use in young people (aged 15–16) in 2020. Marijuana use usually begins during adolescence [33,34], and early onset of consumption increases the risk of developing a substance use disorder later in adulthood [31,35,36,37]. It is well known that adolescents are more vulnerable to drugs because their limbic system is fully formed while their cortical areas responsible for decision making are still developing [38,39]. This innate instinct to try new experiences together with the misperception of cannabis use safety created by all the laws changing worldwide can pose a real global public health threat. Moreover, e-cigarettes and other electronic nicotine-delivery systems, marketed as “safer alternatives” to avoid combustion [40,41], represent new tools for adolescents to smoke cannabis by simply buying devices and liquids on the Internet. Notably, in 2019, in the US an outbreak of severe lung injury was reported after the use of unlicensed vaping products containing THC, CBD, and nicotine [38,42,43,44,45,46].

THC and CBD show completely different effects on the central nervous system, and while THC has been labeled as dangerous because of its harmful psychoactive effects, CBD has been considered “safe” because of its ability to counteract the effects induced by THC. Since several products containing both of these compounds have been marketed in the last decade, people, including adolescents, are consuming CBD by itself. However, there is a lack of information on the long-term effects of CBD exposure; thus, the aim of this review was to collect preclinical and clinical information on the central effects of CBD and to compare them with those induced by THC, focusing in particular on CBD’s rewarding and neuroinflammatory properties, its impact on memory and attention, as well as on substance use and neuropsychiatric disorders.

### 1.3. THC and CBD Pharmacology

As previously mentioned, THC and CBD act on the endocannabinoid system (eCBS), which plays important roles in central nervous system (CNS) development and synaptic plasticity and participates in several physiological processes (e.g., motor control, pain perception, regulation of energy balance, and the immune system) [47,48,49,50,51,52].

The eCBS is a complex neuromodulatory system that includes the endogenous cannabinoids (eCBs), such as anandamide (arachidonoyl ethanolamide, AEA) and 2-arachidonoyl glycerol (2-AG) [53,54], and all the proteins that transport, synthesize, and degrade eCBs, such as N-acylphosphatidylethanolamine (NAPE)-specific phospholipase d-like hydrolase (NAPE-PLD), fatty acid amide hydrolase (FAAH), diacylglycerol lipase α (DAGLα) and DAGLβ, fatty acid amide hydrolase 1 (FAAH), and monoacylglycerol lipase (MAGL) [55,56,57] (for an overview of the eCBS, see Figure 3).

Although both THC and CBD mainly interact with metabotropic CB1 and CB2, THC binds both with nanomolar affinity, while CBD displays micromolar affinity [65,66,67,68]; additionally, they exert completely different actions on these two receptors (see Figure 4). Indeed, while THC acts as a partial agonist [69], CBD exerts a negative allosteric modulatory action on CBRs [70], reducing the potency and efficacy of CBR agonists, such as THC, but also of the endogenous eCB ligands, e.g., AEA and 2-AG [71,72]. Through complex mechanisms of action, both molecules can also modulate eCBS signaling (e.g., eCBS reuptake proteins and enzymes) and influence eCB levels, for instance, by increasing AEA levels [72].

Besides their “classical” actions on CBRs, recent evidence shows that THC and CBD can also interact, in different ways, with other receptors activated by eCBs (see Figure 4); in particular, THC acts as an agonist of the peroxisome proliferator-activated alfa and gamma (PPARα-γ) receptors, orphan G-protein coupled receptors 55 and 18 (GPR55, GPR18), and transient receptor potential of vanilloid 2-4 (TRPV2-4) and ankyrin (TRPA1) channels and as an antagonist of the TRP cation channel subfamily M member 8 (TRPM8) and serotonin 3A receptor (5-HT3A) [73,74,75,76,77]. On the other hand, CBD acts as an agonist of the receptors/channels TRPA1, TRPV1-3, PPARγ, 5-HT1A, and A2 and A1 adenosine and as an antagonist of the receptors GPR55, GPR18, and 5-HT3A. CBD is also an inverse agonist of the receptors GPR3, GPR6, and GPR12 [78,79]. Moreover, activity at the μ and δ opioid receptors has also been reported [69,74,80].

Thus, the specific biological/pharmacological effects of THC and CBD are most likely due to their pharmacological promiscuity rather than merely to CBR activity. For instance, preclinical evidence demonstrated that CBD-induced anxiolytic effects depend on its 5-HT1A activity [81,82]. Similarly, multiple-receptor mechanisms seem to be implicated in anti-psychotic CBD effects (see review [83]), as well as in analgesic, anti-inflammatory, and cognitive effects induced by these two phytocannabinoids [74,84]. Finally, important roles for CB1R heteromers (e.g., CB1R-5-HT2AR and A2AR-CB1R) in cognitive effects induced by THC and CBD have recently been demonstrated [85,86].

Although these and other findings suggest a key role of non-CB receptors in addictive behaviors (see Section 2.1), the specific influences of each receptor/channel on other THC/CBD effects (e.g., analgesic, neuroinflammatory, and cognitive effects) are still unknown.

Therefore, given their wide pharmacological targets, further studies are necessary to understand the precise mechanisms underlying CBD and THC actions, and subsequently their potential therapeutic properties.

## 2. Divergent Central Effects of THC and CBD

### 2.1. Reward and Substance Use Disorders (SUDs)

Some of the major differences between THC and CBD are their highly divergent central effects. Indeed, through different preclinical experimental approaches (i.e., conditioned place preference and electrical brain stimulation), it has been demonstrated that CBD, unlike THC, does not show any rewarding effects [87,88,89] or psychoactive properties [90,91,92]; this may be due, in part, to its inability to alter extracellular dopamine (DA) levels in the ventral striatum [83]. Moreover, whereas THC exposure is able to induce dysregulation of mesolimbic DA transmission and affect salience stimuli evaluation [93,94,95,96], CBD is able to normalize/restore aberrant DA signaling and salience processing [93,94,95,96]. These opposite effects are due to the different pharmacological properties shown by these two cannabinoids. Indeed, as mentioned before, THC is a partial agonist of CB1 and CB2 receptors, whereas CBD acts as a negative allosteric modulator of both CBRs [97,98,99,100,101], reducing the potency and efficacy of CB1R agonists [97,102,103]. This diametrically opposite action of CBD on CB1Rs may represent the principal mechanism in attenuating the psychoactive adverse effects of THC [104]. Moreover, CBD is a partial agonist of D2 receptors, inhibits FAAH, and stimulates TRPV1 and 5-HT1A receptors [81,97,105], which may play an important role in CBD’s action. Little is known about CBD’s effects upon the mesolimbic system and DAergic function, and preclinical studies suggest that CBD may inhibit FAAH, resulting in an increase in AEA signaling [105] which can block DA release at the presynaptic level [106]. Seeman, in 2016 [107], proposed that CBD’s antipsychotic effects were mediated by its action as a partial agonist of DA receptors, while other authors have suggested that CBD can antagonize the effects of THC on DAergic function, mitigating many of the psychotropic side effects of THC [83,108,109,110]. Besides counteracting THC effects, CBD seems to represent a therapeutic strategy against different substance use disorders (SUDs). Growing preclinical evidence obtained in adult rodents with different experimental paradigms (i.e., intravenous drug self-administration, conditioned place preference, and intracranial brain-stimulation reward) has demonstrated that CBD is able to reduce craving, withdrawal symptoms, and relapse induced by different drugs of abuse (e.g., cocaine, heroin, amphetamine, and ethanol) (see the review of Galaj 2020) [83]. Conversely, only a few clinical studies have been conducted and only some have shown the efficacy of CBD in SUDs. Indeed, CBD did not affect tobacco withdrawal [111], but it was able to attenuate opioid subjective-cue-induced craving for up to 7 days after the end of treatment [112]. In adult cannabis-dependent subjects, reductions in craving and withdrawal symptoms were reported only with different dosages of *Nabiximols*, a combination of CBD and THC [111,112,113,114,115,116,117,118]. In adolescents, although no preclinical studies concerning the effect of CBD in SUDs were identified, two case reports (of a 19-year-old woman with cannabis dependence and of a 16.9-year-old man with multiple-substance use disorder, such as cannabis, cocaine, and ecstasy use) described the efficacy of CBD in reducing craving and withdrawal symptoms [115,116]. Despite the main targets of CBD being the CBRs, as previously mentioned, in vivo and in vitro preclinical studies indicate that CBD may also act on other receptors/channels (e.g., 5-HT1A, A1, GPR55, PPARγ, TRPV1, and μ and δ opioid receptors) in physiologic conditions [69,74,75,76,77,78]. Therefore, it is plausible to hypothesize that CBD, through these multiple-receptor mechanisms, normalizes DA transmission altered by drugs of abuse, reducing drug taking, seeking, and relapse [83,119]. Therefore, further studies are required to understand the role of CBD as a potential pharmacotherapy for SUDs in adolescence. Moreover, while the rewarding and addictive properties of THC have been widely evaluated in both youths and adults [39,95,96,120], little is known about the rewarding effects of CBD in adolescence, both at clinical and preclinical levels. It is well-known that in humans cannabinoid exposure (high THC/low CBD concentrations) during adolescence is associated with a higher risk (four times as compared to adult exposure) of developing cannabis dependence [33], and the risk increases with THC content [121]. This adolescent sensitivity to cannabinoid exposure is due to the critical role of the eCBS, which plays a central modulatory role in regulating the neurodevelopment of reward and stress circuitry in the brain [112,122,123]. Therefore, the exposure to THC during adolescence induces changes in neurodevelopmental trajectories, leading to long-lasting effects, such as vulnerability to drug addiction and psychotic episodes [124]. Several preclinical and epidemiological studies suggest that adolescent THC exposure may predispose to the abuse of other illicit drugs (i.e., cocaine, heroin, and amphetamines) in later adulthood, thus promoting drug dependence (the so-called “Gateway Hypothesis”) [125,126,127,128,129,130,131]. For example, chronic THC exposure during adolescence increased opioid self-administration and opioid CPP in adult male rats [132,133,134]. Although multiple studies support the Gateway Hypothesis—a causal relation between adolescent THC exposure and development of different SUDs later in life—numerous pieces of evidence have failed to provide support for this theory. In particular, genetic epidemiological and preclinical studies have shown that there are some common genetic risk factors shared by all drugs of abuse that are the basis for the association between the use of cannabis and other illicit drugs [39,129,132,135,136]. Besides genetic epidemiological studies, other preclinical research does not support this causality, suggesting, conversely, that adolescent THC pre-exposure actually reduces sensitivity to the rewarding effects of other illicit drugs, such as heroin [133,137,138].

To the best of our knowledge, there is a lack of material in the literature on the rewarding and addictive effects of CBD exposure during adolescence, as well as on its potential relation to other drugs of abuse. For instance, Klein and colleagues (2011) demonstrate that CBD potentiates the psychoactive and physiological effects of THC in adolescent rats, most likely acting on its metabolism [139].

Therefore, although preclinical and clinical evidence obtained in adults suggests that CBD is “safe”, it cannot be ruled out that there are possible addictive effects induced by CBD exposure during adolescence.

### 2.2. Neuropsychiatric Disorders

A wide range of phytochemicals possesses biological activities affecting a variety of neurological and neuropsychiatric disorders [140].

The eCBS plays an important role in brain development processes, such as neuromaturation, synaptic pruning, myelination, and receptor distribution, that occur during adolescence [141,142,143,144,145,146].

Therefore, the over-stimulation of the eCBS induced by chronic exposure to CB1R agonists during this critical period can dramatically affect neurodevelopment, causing long-lasting consequences, ranging from emotional and cognitive deficits to neuropsychiatric symptoms [147,148,149,150]. Importantly, the currently available formulations of cannabis (high-potency strains) and synthetic cannabinoids (full CBR agonists) may influence brain development even more, leading to worse outcomes compared to those reported in past generations of cannabinoid users [151,152].

Decades of clinical evidence demonstrate that early-onset marijuana use (before 17 years old) is associated with a higher risk of developing neuropsychiatric disorders, such as depression, anxiety, and schizophrenia [131,148,153,154,155,156,157,158].

For instance, a 25% prevalence of depressive disorders has been reported among chronic cannabis users [159,160], and the numbers are even more concerning in women and in early-onset cannabis users [161,162]. Furthermore, the results of a meta-analysis published in 2019 showed that cannabis use in adolescence is associated with a higher likelihood of developing depression and suicide ideation in young adulthood [148]. Moreover, it has been shown that the risk of developing anxiety disorders later in life is doubled in chronic adolescent cannabis users (onset < 15 years old), especially in girls [154,156]. Moreover, persistent depersonalization and amotivational syndromes have been observed sometimes in cannabis users [163].

However, some findings suggest no causal link between early cannabis consumption and likelihood of neuropsychiatric disorders in adulthood [164,165]. Genetic predispositions and environmental factors (e.g., family history, socio-cultural-economic status, and past life-experiences), as well poly-abuse, represent confounding factors that limit the causality hypothesis and overall data understanding. Controversial data have also been observed in animal models of long-lasting consequences of adolescent THC exposure due to discrepancies in doses, routes of administration, and periods of exposure; however, the majority of preclinical data confirm the neuropsychiatric side effects of cannabis (see reviews [166,167]). Importantly, clinical evidence suggests that the magnitude of disorders is positively correlated with the frequency, dose, age at onset of consumption, and THC content [168]. Focusing on this latter aspect, it has been demonstrated that cannabis strains and extracts containing high THC and low CBD concentrations are linked to increased neuropsychiatric risk, highlighting the role of CBD in mitigating THC-related neuropsychiatric side effects [169,170,171,172]. For instance, Hutten and colleagues (2022) recently demonstrated in a placebo-controlled, randomized, within-subjects study that the ability of CBD to counteract THC-induced anxiety depends on THC:CBD ratios [173]. Conversely, a recent double-blind, within-subjects, randomized study (using different THC:CBD ratios) reported no evidence that CBD is able to protect against the acute adverse effects of THC on cognition and mental health [174]. (See Table 1 for a summary of all the clinical studies evaluating THC and CBD effects.)

Besides the studies concerning THC effects, cumulative clinical evidence has demonstrated the anxiolytic and antipsychotic properties of CBD in adulthood (see review [175]).

Clinical studies (i.e., placebo-controlled, case–control studies) have shown that CBD is able to decrease social anxiety symptoms and sedation [176] and anxiety and cognitive impairment during speech performance [177], as well as symptoms (i.e., anxiety and cognitive impairments) in post-traumatic stress disorder (PTSD) patients [178,179,180,181]. A recent clinical trial reported significant improvements in anxiety, mood, sleep, and executive functions in patients with moderate-severe anxiety treated with high-CBD/low-THC sublingual solutions (CBD: 9.97 mg/mL, THC: 0.23 mg/mL, 4 weeks). Although no intoxication or serious adverse events were observed, minor side effects, such as sleepiness/fatigue, increased energy, and dry mouth, were reported [182]. Brain imaging studies suggest that CBD anxiolytic effects could be due to its ability to decrease amygdala activation [183], even though a recent study reported that a single CBD administration (600 mg) did not modify brain responses to emotional faces, cognitive measures of emotional processing, or anxiety [184].

However, multiple lines of clinical evidence have indicated that CBD was not effective in treating anxiety.

For instance, pretreatment with CBD did not improve the outcomes of therapy sessions in a relatively large group of patients diagnosed with anxiety (i.e., social anxiety disorder or panic disorder with agoraphobia) [185]. Moreover, in a randomized study conducted on healthy college students, oral CBD dose administration (150, 300, or 600 mg) did not show anxiolytic effects on test anxiety [186].

Although few clinical studies have evaluated CBD’s antipsychotic properties and its therapeutic efficacy in adults with schizophrenia [187,188,189,190], the majority have shown beneficial effects in reducing psychotic symptoms. For instance, a randomized clinical study demonstrated that, compared to the placebo group, schizophrenia patients treated with CBD (1000 mg/day for 6 weeks) showed fewer positive psychotic symptoms and improvement in cognitive performance and overall functioning, without adverse events [187]. Conversely, Boggs and colleagues (2018) showed that treating antipsychotic-treated schizophrenia patients with CBD for 6 weeks (600 mg/day) had no effect on psychotic symptoms [190]. Finally, a randomized clinical trial comparing CBD (200–800 mg/day) for 4 weeks with amisulpride (an atypical antipsychotic dopamine receptor antagonist) in acute schizophrenia patients showed that both treatments were safe and effective; however, CBD patients displayed fewer side effects (no extrapyramidal symptoms, lower prolactin levels, and less weight gain) [188]. Interestingly, this study pointed out the positive correlation between AEA levels and clinical improvements, suggesting that the inhibition of AEA degradation induced by CBD may be implicated in its antipsychotic effects, representing a new approach in treating schizophrenia. However, these promising results were obtained for small populations (less than 100), with a short follow-up window, or from individual case reports [187,188,190]. Indeed, several case reports also revealed toxic side effects of CBD [191] and increases in anxiety, depressive symptoms, and suicidal ideation [192,193]. Interestingly, it has been demonstrated that the intoxication effects (i.e., dissociative states) induced by CBD may depend on individual cannabinoid history [194,195]. These opposing results might be due to differences in CBD products, from pure CBD to unknown or unclear THC:CBD ratios, and the concentrations used in clinical studies and case reports, respectively. Despite controlled experimental settings (dose, modality, and composition), preclinical studies have not yet resolved the controversy about the consequences and the therapeutic potential of CBD. For instance, CBD administration in adult rodents reduced anxiety-related behaviors, decreasing neuronal activity (i.e., c-fos positive cells) [196,197,198] and cerebral blood flow in brain areas involved in anxiety symptoms, such as the amygdala and cingulate cortex [199]. Repeated CBD exposure was able to prevent long-lasting anxiogenic effects in an animal PTSD model, probably by acting on 5HT1A receptors [177]. Moreover, CBD antidepressant-like effects have been reported in a chronic mild stress mouse model [200]. Finally, in an animal model of schizophrenia, CBD was able to improve psychotic symptoms and decrease stereotypy induced by DA agonists, without catatonia, as observed with conventional treatment (clozapine and haloperidol) [201,202,203]. In a very recent study, Huffstetler and colleagues (2023) demonstrated that a single CBD administration (10 or 20 mg/kg, i.p.) induced changes in mouse behavior in a dose-, sex-, and anxiety-state-dependent manner. Interestingly, CBD decreased anxiety-like behavior in wild-type mice, while it enhanced it in mutant mice (Kv1.3^-/-^) with traits including anxiety-like and attention-deficit-like behaviors [204].

However, the CBD effects on THC-induced anxiety and psychotic symptoms observed in humans and rodents are controversial. In rats, when CBD was added to THC, increased hypomotility was observed, along with other depressive symptoms (decrease in food and water intake) [205]. CBD potentiated rather than inhibited the anxiogenic effects of THC in rats treated chronically (THC:CBD ratio 1:1) [139]. On the other hand, CBD attenuated the reduction in social interaction induced by THC [206]. Another study reported that infralimbic CBD administration was able to induce either anxiogenic or anxiolytic effect in rats depending on the behavioral test performed [82].

All these data were obtained in adult rodents, while only a few preclinical studies have evaluated the impact of CBD and CBD-THC exposure during adolescence.

Interestingly, adolescent CBD exposure in rats potentiated an increase in anxiogenic effects and decreased social interaction induced by chronic THC [139]. Recently, Kasten and colleagues (2019) demonstrated that acute CBD-THC administration induces behavioral deficits, such as increased anxiety-like behaviors. Moreover, repeated CBD-THC exposure during adolescence induced pronounced, long-lasting effects in female but not male rats [207].

Unfortunately, not many clinical studies have evaluated either the impact of adolescent CBD exposure on the development of neuropsychiatric disorders or its possible protective effects [208,209]. Masataka (2019) reported that repeated CBD treatment (300 mg/kg, 4 weeks) significantly decreased anxiety in teenagers with social anxiety disorders. However, no systematic evaluation of side effects was conducted [208]. Consistently, an open-lab study reported that CBD (up to 800 mg/day for 12 weeks) could reduce anxiety severity without serious adverse effects in young people (12–25 y) with treatment-resistant anxiety disorders. Although CBD treatment did not induce serious adverse effects, 80.6% of patients reported fatigue, low mood, cold chills, and hot flushes [209]. Therefore, caution regarding CBD consumption in youth is warranted.

In summary, while the long-term effects induced by adolescent THC exposure have been extensively characterized, studies evaluating the consequences of CBD exposure are lacking. One of the major concerns is the inconsistency in CBD and THC contents among commercial products, due to the lack of rigorous lab monitoring and legal regulation. Clinical controlled-setting studies (dose, composition, and timing) with larger sample sizes and longer follow-up periods are necessary to deeply understand the toxic and therapeutic properties of CBD in youth. Importantly, a randomized controlled study is evaluating for the first time the efficacy of CBD treatment in youths (12–25 y) with ultra-high risk for psychosis [210]. Therefore, so far, it is dangerous to consider CBD a “safe and beneficial drug”.

**Table 1 ijms-24-05251-t001:** Clinical trials evaluating the effects of THC and CBD on neuropsychiatric disorders in adults (white rows) and adolescents (light-grey rows).

Experimental Design	Doses	Results	Safety, Compliance, and Side Effects	References
Case report describing 6 patients (5 male; 1 female; ages 33, 24, 16, 19, 16, and 18) who developed persistent depersonalization disorder in adolescence after consuming cannabis.	Patients smoked their own cannabis	All reported cases described onset of depersonalization disorder in adolescence. In 2 of these cases, the illness course was severely disabling.	N/A	[163]
Participants divided into case group (cannabis users with a first episode of psychosis; n = 280, 18–65 y) and control group (healthy patients; n = 174, 18–65 y) were assessed for sociodemographic data and use of illicit drugs, including cannabis.	Patients smoked their own cannabis	Patients in the case group were more likely to be current daily users and to have smoked cannabis for more than 5 years. Among those who used cannabis, 78% of the case group used high-potency cannabis (sinsemilla, ‘skunk’) compared with 37% of the control group.	N/A	[170]
A randomized, double-blind, between-subjects design trial (n = 48 participants with previous cannabis use, 21–50 y). All participants were assessed at three separate time-points: (1) baseline; (2) post-CBD; and (3) post-THC. All participants were assessed for traits of paranoia, cannabis dependence, psychotic/dysphoric experiences following recreation cannabis use, positive psychotic dimension, mood, and cognitive functioning.	Capsule with CBD 600 mg (n = 22) or placebo (n = 26) 210 min ahead of intravenous THC (1.5 mg)	Clinically significant positive psychotic symptoms were less likely in the CBD group compared with the placebo group. In agreement, post-THC paranoia was less common in the CBD group compared with the placebo group. Episodic memory was poorer relative to baseline in the placebo pre-treated group compared with the CBD group.	N/A	[171]
A double-blind, placebo-controlled, within-subjects study (n = 26 healthy occasional cannabis users, 10 males and 16 females, mean age 23.1 y) with 4 treatment conditions separated by a minimum washout period of 7 days to avoid potential carry-over effects. The order of treatment conditions was randomized across participants. All participants were assessed for anxiety, pain, and emotional state. All drugs were self-administered by vaporisation at 200 °C.	THC-dominant cannabis (13.75 mg THC, THC 22%, and CBD < 1%), CBD-dominant cannabis (13.75 mg CBD, THC < 1% and CBD 9%), THC/CBD-equivalent cannabis (13.75 mg THC/13.75 mg CBD), or cannabis placebo (<0.2% total cannabinoid content)	Both THC and THC/CBD significantly increased self-rated state anxiety compared to placebo. State anxiety after THC/CBD was significantly lower than after THC alone. THC-induced anxiety was independent of anxiety at baseline. When baseline anxiety was low, CBD completely counteracted THC-induced anxiety; however, when baseline anxiety was high, CBD did not counteract THC-induced anxiety. There were no effects of any treatment condition on emotional state.	N/A	[173]
A double-blind, randomized, four-arm, within-subjects trial in which participants (n = 46 healthy infrequent cannabis users, 21–50 y) inhaled 4 different cannabis vaporized preparations (randomized, counter-balanced order, with minimum one-week wash-out period between each treatment exposure). All participants were assessed for delayed verbal recall, severity of psychotic symptoms, and cognitive, subjective, pleasurable, pharmacological, and physiological effects.	THC 10 mg—CBD 0 mg (0:1 CBD: THC), THC 10 mg—CBD 10 mg (1:1), THC 10 mg—CBD 20 mg (2:1), or THC 10 mg—CBD 30 mg (3:1)	THC (0:1) was associated with impaired delayed verbal recall and induced positive psychotic symptoms. These effects were not significantly modulated by any dose of CBD. Furthermore, there was no evidence of CBD modulating the effects of THC on other cognitive, psychotic, subjective, pleasurable, or physiological measures.	N/A	[174]
A double-blind, within-subjects, placebo-controlled study in university students (n = 10, 20–33 y, naive to treatment) with generalized social anxiety syndrome. All participants were assessed for severity of social phobia disorder and phobia. Regional cerebral blood flow at rest and after treatment was measured twice using Technetium-99m-labeled ethyl cysteinate dimer (ECD) single-photon emission computed tomography (SPECT).	Oral dose of CBD, 400 mg or placebo	Relative to placebo, CBD was associated with significantly decreased subjective anxiety, reduced ECD uptake in the left parahippocampal gyrus, hippocampus, and inferior temporal gyrus, and increased ECD uptake in the right posterior cingulate gyrus.	N/A	[176]
A double-blind randomized, placebo-controlled trial in never-treated patients (mean age 23 y) with SAD (n = 24) and healthy controls (HC, n = 12). At 6 time points during a simulation public speaking test (SPTS), all participants were assessed for mood, negative state, and physiological measures (blood pressure, heart rate, and skin conductance).	CBD 600 mg (n = 12) or placebo (n = 12) 1 h and half the SPTS test. HC (n = 12) participants did not receive any medication	Pretreatment with CBD significantly reduced anxiety, cognitive impairment, and discomfort in speech performance, and significantly decreased alertness in anticipatory speech. The placebo group presented higher anxiety, cognitive impairment, discomfort, and alert levels when compared with the HC. The increase in negative states during the testing observed in the placebo group was almost abolished in the CBD group. No significant differences were observed between CBD and HC in the cognitive impairment, discomfort, and alert factors.	N/A	[177]
A double-blind, placebo-controlled, between-subjects trial in which participants (n = 48, 18–35 y) were randomized to three groups (each n = 16) to receive either (1) CBD prior to extinction (CBD pre-extinction group), (2) CBD following extinction (CBD post-extinction group), or (3) placebo (placebo group). In this study a sub-anxiolytic CBD dose was used. All drugs were vaporized at 210 °C and administered via a Volcano Medic vaporizer. At recall, 48 h later, in the conditioning session, all participants were exposed to conditioned stimuli and conditioning contexts before (recall) and after (reinstatement) exposure to the unconditioned stimulus. Skin conductance and shock expectancy measures of conditioned responding were recorded throughout. All participants were assessed for depressive symptoms, trait anxiety, verbal IQ, and non-emotional explicit memory.	32 mg of inhaled CBD prior to extinction (CBD pre-extinction group), 32 mg of inhaled CBD following extinction (CBD post-extinction group) or placebo	CBD given post-extinction enhanced consolidation of extinction learning as assessed by shock expectancy. CBD administered at either time produced trend-level reduction in reinstatement of autonomic contextual responding. No acute effects of CBD were found on extinction.	N/A	[178]
A double blind, randomized trial in patients diagnosed with PTSD (n = 33 of both sexes, 18–60 y) treated with CBD or placebo. In the first experimental section, all participants were matched by sex, age, body mass index, and PTSD symptoms. On the same day, participants prepared the behavior test, recording accounts of their traumas in digital audio for a minute and a half and then imagining the trauma for 30 s. After 7 days, participants were treated (CBD or placebo) before performing the behavioral test, listening to the trauma account and imagining themselves in that situation. Before and after the behavioral test, all participants were assessed for subjective changes in mood and anxiety, physiological correlates of anxiety (blood pressure, heart rate, and salivary cortisol). Seven days later, participants underwent the same procedures as in the previous session, but without the pharmacological intervention, to assess the effect on reconsolidation of traumatic memories.	CBD 300 mg (n = 17) or placebo (n = 16)	CBD significantly attenuated the cognitive impairment effect that persisted 1 week after drug administration. No significant differences between the effects of CBD and placebo on anxiety, alertness, and discomfort induced by the recall of the traumatic event during the pharmacological intervention and in the subsequent week. There were no significant differences between the CBD and placebo groups regarding physiological data.	N/A	[181]
An open-label stage of clinical trial phase 2 (NCT02548559) autoregressive linear modeling assessed efficacy and tolerability of 4-week treatment with high-CBD sublingual solution in 14 outpatients with moderate-to-severe anxiety. Secondary outcomes: at baseline and week 4, patients were assessed for mood, depressive symptoms, sleep disturbance, sexual function, quality of life, and cognitive functions (battery of cognitive tests).	1 mL t.i.d of high-CBD sublingual solution (CBD 9.97 mg/mL, THC 0.23 mg/mL)	Significant improvement in primary outcomes measuring anxiety and secondary outcomes assessing mood, sleep, quality of life, and cognition (specifically executive function) following treatment. Anxiety was significantly reduced at week 4 relative to baseline. Clinically significant treatment response (≥15% symptom reduction) was achieved and maintained as early as week 1 in most patients; cumulative frequency of treatment responders reached 100% by week 3.	The study drug was well-tolerated, with high adherence/patient retention and no reported intoxication or serious adverse events. Minor side effects, including sleepiness/fatigue, increased energy, and dry mouth, were infrequently endorsed.	[182]
A double-blind, randomized, placebo-controlled study in 15 healthy men (18–35 y) who had used cannabis 15 times or fewer in their life. Regional brain activation (blood-oxygenation-level-dependent response), electro-dermal activity (skin conductance response, SCR), and objective and subjective ratings of anxiety were assessed after treatment. All participants were assessed for cannabis and other illicit substance use and underwent urine drug screen analyses prior to each session. Periodic (at baseline and at 1, 2, and 3 h post-administration) psychopathological ratings of mood, anxiety, intoxication, and psychotic symptoms were collected for all participants.	Gelatin capsule with THC 10 mg, CBD 600 mg, or placebo	THC increased anxiety, as well as levels of intoxication, sedation, and psychotic symptoms, whereas there was a trend for a reduction in anxiety following administration of CBD. The number of SCR fluctuations during the processing of intensely fearful faces increased following administration of THC but decreased following administration of CBD. CBD attenuated blood oxygenation-level-dependent signaling in the amygdala and the anterior and posterior cingulate cortex while subjects were processing intensely fearful faces, and its suppression of the amygdalar and anterior cingulate responses was correlated with a concurrent reduction in SCR fluctuations. THC mainly modulated activation in frontal and parietal areas.	No serious adverse events (deaths, hospitalizations, or emergency department visits) occurred during the study. Three subjects from the original samples (n = 18) had a psychotic reaction to THC administration and were excluded since they were unable to perform the tests (final sample, n = 15). These subjects were followed up for 24 h until the psychotic symptoms relieved. They were further monitored monthly and remained well, with no psychiatric or clinical symptoms.	[183]
A double-blind, randomized, placebo-controlled crossover study of acute oral challenge of CBD in 24 healthy participants (12 male, 12 female, 18–70 y) on emotional processing, with neuroimaging (viewing emotional faces during functional magnetic resonance imaging) and cognitive (emotional appraisal) measures, as well as subjective response to experimentally induced anxiety.	CBD 600 mg or placebo	CBD did not produce effects on brain responses to emotional faces and cognitive measures of emotional processing or modulate experimentally induced anxiety relative to placebo.	N/A	[184]
A double-blind, randomized, placebo-controlled trial in patients (n = 80, 18–65 y) with panic disorder with agoraphobia or social anxiety disorder. All participants were exposed to 8 therapist-assisted exposure in vivo sessions (weekly, outpatient) under the treatment condition. The Fear Questionnaire (FQ) was assessed at baseline, mid- and post-treatment, and at 3- and 6-month follow-ups.	Oral CBD 300 mg (n = 39) or placebo (n = 41)	No differences were found in treatment outcomes over time between CBD and placebo groups in terms of FQ scores.	Incidence of adverse effects was equal in the CBD and placebo conditions.	[185]
A double-blind parallel, randomized, placebo-controlled study in healthy college students (n = 32) who self-reported moderate-to-severe levels of test anxiety (TA). This study tested single oral-administration doses of CBD, compared to placebo, for reducing test anxiety (TA) in a researcher-derived experimental analog. After treatment, all participants completed a statistics examination, and measures of TA and general anxiety were assessed during examination administration.	CBD (150, 300, or 600 mg) or placebo	No effect of CBD dose on self-reported TA or general anxiety.	N/A	[186]
An exploratory double-blind, randomized, parallel group, placebo-controlled trial in patients (n = 88, 18–65 y) with schizophrenia or related psychotic disorders. Patients were randomized to receive CBD or placebo alongside their existing antipsychotic medication. All participants were assessed before and after treatment for mood, psychotic symptoms, cognitive functions, and improvement in clinical state.	CBD 1000 mg/day (n = 43) or placebo (n = 45) for 6 weeks	After 6 weeks of treatment, compared with the placebo group, the CBD group had lower levels of positive psychotic symptoms and were more likely to have been rated as improved and as not severely unwell by the treating clinician. Patients who received CBD also showed greater improvements in cognitive performance and in overall functioning.	CBD was well-tolerated, and rates of adverse events were similar between the CBD and placebo groups.	[187]
A parallel group, randomized, placebo-controlled study in stable antipsychotic-treated patients (n = 36, 18–65 y) diagnosed with chronic schizophrenia. Patients were randomized to receive CBD or placebo augmentation. All participants were assessed for cognitive functions (at baseline and at end of 6 weeks of treatment) and psychotic symptoms (at baseline and biweekly).	CBD 600 mg/day or placebo for 6 weeks	CBD treatment was ineffective on psychotic symptoms and on cognitive functioning.	Side effects were similar between CBD and placebo, with the one exception being sedation, which was more prevalent in the CBD group.	[190]
A therapeutic exploratory (phase II, NCT00628290), double-blinded, monocenter, randomized, parallel-group, controlled clinical trial of CBD vs. amisulpride efficacy in patients (n = 39, 18–59 y) with diagnosis of schizophrenia or schizophreniform psychosis. All participants were assessed for psychotic symptoms, and measurements of serum prolactin and body weight were taken. Safety measures included repeated electrocardiograms as well as routine blood parameters.	CBD (200–800 mg/day) or amisulpride (200–800 mg/day), 28 days of treatment	Patients undergoing either CBD or amisulpride treatment showed significant clinical improvement (reduction in psychotic and other symptoms of schizophrenia). No significant differences in the clinical effects between treatments were observed.	Both treatments were safe and led to significant clinical improvement, but CBD displayed a markedly superior side-effect profile. Compared with amisulpride, CBD was associated with significantly fewer extrapyramidal symptoms, less weight gain, and lower prolactin increase. Furthermore, CBD was well-tolerated and did not significantly affect hepatic or cardiac functions.	[188]
A randomized, placebo-controlled trial to examine the acute effects of THC and CBD alone and in combination in frequent and infrequent cannabis users. Thirty-six participants (31 male, 18–51 y) were subsequently divided into groups of frequent cannabis users (n = 18, 17 male, 21–44 y) and infrequent users/non-naive nonusers (n = 18, 14 male, 18–51 y). All participants were objectively and subjectively assessed for intoxication (primary outcomes). Additional indices of intoxication were assessed (psychiatric symptoms, depression, and anxiety).	THC (8 mg), high CBD (400 mg), THC + low CBD (THC: 8 mg, CBD: 4 mg), THC + high-CBD (THC: 12 mg; CBD: 400 mg) or placebo (ethanol vehicle 400 μL). Five vaporization sessions, with a 1-week washout between	CBD showed some intoxicating properties relative to placebo. Both frequent and infrequent users subjectively reported feeling intoxicated by high-dose CBD administered alone (i.e., not combined with THC), with protracted effects across the 3 h session relative to placebo, but this was not corroborated by the objective intoxication measure. Low doses of CBD when combined with THC enhanced, while high doses of CBD reduced the intoxicating effects of THC. The enhancement of intoxication by low-dose CBD was particularly prominent in infrequent cannabis users and was consistent across objective and subjective measures.	See results	[195]
An open-label trial, in young people (n = 31, 12–25 y) with anxiety disorder and no clinical improvement despite treatment with cognitive–behavioral therapy and/or antidepressant medication. All participants received additional CBD treatment. The primary outcome was improvement in anxiety severity at week 12. Secondary outcomes included comorbid depressive symptoms and social and occupational functioning.	CBD treatment on a fixed–flexible schedule (titrated up to 800 mg/d) for 12 weeks	CBD decreased anxiety from baseline to week 12 (−42.6%). Depressive symptoms and functioning improved significantly.	Adverse events were reported in 25 (80.6%) of 31 participants and included fatigue, low mood, and hot flushes or cold chills. There were no serious and/or unexpected adverse events.	[209]
A 21 y longitudinal study of a birth cohort (New Zeland). Participants were annually assessed for frequency of cannabis use (from 14 to 21 y), and for psychosocial outcomes including property/violent crime, depression, suicidal ideation, suicide attempt, and other illicit drug use.	Patients smoked their own cannabis	Association between frequency of cannabis use and all outcomes, particularly other illicit drug use. Age-related variation in the strength of association between cannabis use and crime, suicidal behaviors, and other illicit drug use, with younger (14–15 y) users being more affected by regular cannabis use than older (20–21 y) regular users. Association between cannabis use and depression did not vary with age.	N/A	[153]
A 6 y cohort study (7 wave) (Australia); 1601 students (14–15 y). Participants were assessed for measure of depression and anxiety at wave 7 (age 21 y).	Patients smoked their own cannabis	60% of participants had used cannabis by the age of 20; 7% were daily users at that point. Daily use in young women was associated with a more than 5-fold increase in the odds of reporting of depression and anxiety. Weekly or more frequent cannabis use in teenagers predicted approximately 2-fold increase in risk for later depression and anxiety. Depression and anxiety in teenagers predicted neither later weekly nor daily cannabis use.	N/A	[154]
A 21 y longitudinal study of a birth cohort (n= 3239) (Australian). All participants were interviewed to assess depression and anxiety using at age 14 y and at age 21 y.	Patients smoked their own cannabis	Those who used cannabis before age 15 y and used it frequently at 21 y were more likely to report symptoms of anxiety and depression in early adulthood. This association was of similar magnitude for those who had only used cannabis and those who reported having used cannabis and other illicit drugs.	N/A	[156]
A logistic regression analysis of data from the 1992 NLAES study (n = 42,862 young adults, 18–29 y) (USA). Participants were assessed for drug dependence, depression, and sociodemographic factors.	Patients smoked their own cannabis	The risk of cannabis abuse and dependence was found to increase with the frequency of smoking occasions and slightly decreased with age. More severe comorbidity was associated with dependence compared to abuse, suggesting that cannabis might be used to self-medicate major depression. The strength of the association between cannabis use and abuse was also increased as a function of the number of joints smoked among females, but not males. With respect to cannabis abuse, the odds for abuse were approximately 2 times greater among males than females. The odds of dependence were 2.6 times greater among those respondents with comorbid major depression, 2.2 times greater among respondents with a comorbid drug use disorder, and 2.7 times greater among respondents with comorbid alcohol dependence compared to those not so classified. Sex was found to modify the use ± abuse relationship—the number of joints smoked per smoking occasion increased the risk for abuse, but only among females. The odds of abuse were 2.4 times greater among females who smoked on average two joints per occasion compared to those who smoked 0.50 joints on a typical occasion. For females who smoked on average eight joints per occasion, the odds of cannabis abuse were 5.5 times greater relative to the odds of smoking joints per occasion.	N/A	[161]
A web-based cross-sectional study on cannabis use and subclinical psychiatric experiences using the Community Assessment of Psychic Experiences; n = 1877 Dutch young adults and adolescents (18–25 y) consuming the same type of cannabis on the majority of occasions (60% of occasions).	THC and CBD exposure were estimated based on Trimbos Institute annual report on Dutch market	Significant inverse relationship between CBD content and self-reported positive symptoms, but not with negative symptoms of depression.	N/A	[172]
A double-blind, placebo-controlled study to assess the efficacy of CBD treatment in Japanese late teenagers (n = 37, 18–19 y) with social anxiety disorder (SAD). Cannabis oil containing CBD or placebo daily for 4 weeks. All participants were assessed for SAD symptoms at the beginning and end of the treatment period.	Cannabis oil containing 300 mg CBD (n = 17) or placebo (n = 20), for 4 weeks	CBD significantly decreased anxiety measured by both scales.	None of the participants had any significant health complaints, although no systematic evaluation of side effects was conducted.	[208]

### 2.3. Memory and Attention

Besides favoring neuropsychiatric disorders, THC exposure may induce deficits in cognitive functions. Common side effects observed both in humans and in animals after THC exposure (acute and chronic) are cognitive impairments, such as disruptions to working memory and attentional and learning deficits, which are positively correlated with THC concentration and negatively correlated with age of exposure [211]. Indeed, clinical evidence demonstrated that these deficits are stronger in early age at onset (adolescent) and in heavy use with high THC:CBD ratio preparations (see review [212]). Recently, imaging studies confirmed an association between cannabis consumption and altered activation patterns during different memory tasks [213]. Consistently, the age of cannabis users affects human brain function, as observed by comparing adolescents, adults, and healthy controls [214]. Accordingly, while chronic exposure to low doses of THC restores cognitive functions, such as memory deficits and learning capacity in old mice (12–18 months), in young adult mice (2 months old) the same THC dose induces cognitive deficits [215]. Moreover, in rodents, THC exposure in adolescence induces memory impairment in the novel object recognition (NOR) test in adulthood [216,217,218,219,220,221,222,223], while milder or non-significant deficits have been observed with THC exposure in adulthood [223,224,225,226]. Taken together, this evidence suggests that THC-induced cognitive impairments may be due to its effects on the adolescent brain. In particular, in affecting the maturation of the brain, such as the cortical areas, THC may lead to memory and cognitive alterations/deficits in adulthood. Besides age of onset, THC:CBD concentration represents a critical factor in determining the cognitive consequences of cannabis consumption. For instance, clinical trials (see Table 2) demonstrated that consumption of cannabis with a low CBD content is associated with memory impairment [227,228], and the psychotic effects are directly proportional to THC content [229]. On the contrary, CBD administration (200 mg/day, 10 weeks) in daily cannabis users induced an improvement in memory functions, such as verbal learning and attentional switching [230]. Consistently, naturalistic studies conducted in regular cannabis users demonstrated that greater CBD concentrations lead to better memory performances [228]. However, a randomized double-blind trial reported that co-administration of a medium dose of CBD (1:2 THC:CBD ratio, 8 and 16 mg, respectively) did not attenuate the memory impairment induced by vaporized THC [231]. The same study reported no cognitive effects when 16 mg of CBD was administered alone. In summary, multiple clinical trials have shown no negative effects of CBD on working memory [232,233], though it is still unclear whether using high-CBD cannabis is enough to counteract the memory impairment induced by THC [228,229,232]. Controversial results have been observed also in animal models. For instance, in rhesus macaque monkeys, CBD pre-treatment (30 mg/kg, i.m., 60 min prior) inhibited the reduction in food seeking induced by 0.3 mg/kg but not 1.0 mg/kg of THC [234]. In the same study, the administration of 30 mg/kg but not 10 mg/kg of CBD alone induced performance impairment. In 2015, Taffe and colleagues reported that CBD is able to ameliorate or reverse some THC-induced impairment of bimanual motor coordination [235]. Interestingly, an improvement in object spatial memory tasks was observed in macaque monkeys when CBD and THC were administered in equal amounts (0.5 mg/kg, i.m.) [236]; however, the THC:CBD ratio in street cannabis is usually very different from 1:1 [227,237]. CBD reduced THC-induced deficits in go-trial success in a stop-signal task in male macaque monkeys when administered at a 1:3 THC:CBD ratio [238]. Differently, data obtained in rodents did not confirm the beneficial effects of CBD on memory observed in non-human primates.

For instance, in rats, CBD (50 mg/kg) did not produce deficits in spatial working memory but did not reverse the spatial memory deficit induced by high THC concentrations [239]. Differently, Hayakawa and colleagues (2008) reported that the same and lower CBD doses (10 and 50 mg/kg) in mice enhanced the impairment of spatial memory induced by THC (1 mg/kg) [240]. This evidence suggests that rodents may not be the right preclinical models for evaluating the effects of THC and CBD in complex memory tasks.

On the other hand, accumulating and consistent preclinical evidence indicates that CBD is able to regulate emotional memory processing, facilitating the extinction or the disruption of the reconsolidation of fear memories, and most likely also drug memories (see review [241]). These beneficial effects seem to be related to the anxiolytic and anti-stress properties of CBD [82,183], which may be correlated with its 5-HT1A activity. Thus, CBD may represent a potential therapeutic candidate in treating PTSD and phobias, as well as addiction disorders.

To the best of our knowledge, little information exists about the impact of CBD exposure during adolescence on memory functions. Recently, Murphy and colleagues (2017) demonstrated that CBD co-administration during adolescence (3 weeks) prevents THC-induced memory impairment in mice, whereas chronic CBD alone did not induce any effect on NOR performance [217]. In a case report of three patients diagnosed with adult attention-deficit/hyperactivity disorder ADHD aged 18, 22, and 23, introducing cannabis in the treatment regimen ameliorated anxiety, depression, and attention [242]. Although these studies suggest protective/beneficial effects of CBD on cognition, caution with respect to CBD consumption in youth is necessary.

In conclusion, despite some findings obtained in adults suggesting a pro-cognitive effect of CBD, a better preclinical and clinical characterization of CBD is necessary to deeply understand the impact of CBD treatment on memory and learning functions and its potential therapeutic effects, especially in adolescence.

**Table 2 ijms-24-05251-t002:** Clinical trials evaluating the effects of THC and CBD on memory and attention and neuroinflammation.

Experimental Design	Doses	Results	Side Effects	References
A repeated-measures design compared a sample of cannabis users (n = 94, average age: 21) on 2 days: under the influence of the drug (intoxicated day) and when drug-free (drug-free day) approximately 7 days apart. A sample of cannabis was collected from each user and analyzed for levels of cannabinoids. On the basis of the CBD: THC ratios of the cannabis samples, individuals from the top and bottom tertiles were directly compared on indices of the reinforcing effects of drugs, explicit liking, and implicit attentional bias to drug stimuli.	Patients smoked their own cannabis	When intoxicated, smokers of high-CBD: THC strains showed reduced attentional bias to drug and food stimuli compared with smokers of low-CBD: THC strains. Those smoking higher-CBD: THC strains also showed lower self-rated liking of cannabis stimuli on both test days.	N/A	[227]
A repeated-measures design compared a sample of cannabis users (n = 134, average age: 21) assessed on 2 days: under the influence of the drug (intoxicated day) and when drug-free (drug-free day) approximately 7 days apart. A sample of cannabis was collected from each user and analyzed for levels of cannabinoids. On the basis of CBD:THC ratios in the cannabis, individuals from the top and bottom tertiles were directly compared on measures of memory and psychotomimetic symptoms.	Patients smoked their own cannabis	Unlike the marked memory impairment of individuals who smoked cannabis low in cannabidiol, participants smoking cannabis high in cannabidiol showed no memory impairment. Cannabidiol content did not affect psychotomimetic symptoms, which were elevated in both groups when intoxicated.	N/A	[228]
A total of 120 current cannabis smokers (average age: 20), 66 daily users and 54 recreational users, were classified into groups according to the presence or absence of CBD and high versus low levels of THC. All were assessed on measures of psychosis-like symptoms, memory, and depression/anxiety.	Patients smoked their own cannabis	Recreational users showed increased depression, anxiety, and psychosis-like symptoms that were attenuated in those using cannabis containing CBD. Prose recall and source memory were poorer in high-THC strain daily users, while better recognition memory was measured in those using high-CBD strains.	N/A	[229]
A randomized, double-blind crossover design to compare the effects in 48 cannabis users (average age: 21) selected on the basis of (1) schizotypal personality questionnaire scores (low, high) and (2) frequency of cannabis use (light, heavy). The Brief Psychiatric Rating Scale (BPRS), Psychotomimetic States Inventory (PSI), immediate and delayed prose recall (episodic memory), and 1- and 2-back (working memory) were assessed on each day.	Placebo THC 8 mg CBD 16 mg THC 8 mg + CBD 16 mg	THC increased overall scores on the PSI, negative symptoms on the BPRS, and robustly impaired episodic and working memory. Co-administration of CBD did not attenuate these effects. CBD alone reduced PSI scores in light users only. At a ratio of 2:1, CBD does not attenuate the acute psychotic and memory-impairing effects of vaporized THC.	N/A	[231]
Placebo-controlled, double-blind, experimental trial with 60 healthy volunteers. Patients were assessed on working memory, cognitive processing speed, attention, and emotional state.	Placebo THC 20 mg CBD 800 mg THC 20 mg + CBD 800 mg	THC affected performance-related activity and extraversion, reduced cognitive processing speed, and impaired attention performance. Administration of CBD alone did not influence emotional state, cognitive performance, or attention. Interestingly, pre-treatment with CBD did not attenuate the effects induced by THC.	N/A	[232]
Double-blind, placebo-controlled, randomized crossover trial in 39 healthy young subjects (average age: 22). Participants received once a single dose of vaping after learning 15 unrelated nouns. Short-delay verbal memory performance (number of correctly free-recalled nouns) 20 min after learning was assessed.	Placebo CBD e-liquid (5%)	CBD enhanced verbal episodic memory performance and did not have negative impacts on secondary-outcome measures of attention or working-memory performance.	N/A	[233]
Case report describing 3 males (aged 18, 22, and 23) diagnosed with ADHD who integrated cannabis into their treatment regimens.	(Patient1) CBD:THC 20:1, smoking (Patient2) CBD:THC 20:1 oil 1 mL, oral (Patient3) CBD:THC 0:19, smoking	All patients showed substantial improvement in terms of depression, anxiety, and attention.	Short-term memory problems Dry mouth Sleepiness	[242]
HIV-infected, antiretroviral-treated individuals (n = 198 sex = male, age = 45–60) were tested to assess the impact of cannabis use on peripheral immune cell frequency, activation, and function using flow cytometry. Amounts of cannabis metabolites were measured in plasma by mass spectrometry to categorize the subjects into three groups: heavy, medium, and non-cannabis users.	Patients smoked their own cannabis	Heavy cannabis users showed a decrease in frequencies of human leukocyte antigen (HLA)-DR + CD38 + CD4+, CD8+ T-cells, intermediate and nonclassical monocyte subsets, as well as decreased frequencies of interleukin 23- and tumor necrosis factor-α-producing antigen-presenting cells compared to non-cannabis-using individuals.	N/A	[243]
Comparative Study designed to compare the levels of circulating CD16+ monocytes and interferon-γ-inducible protein 10 (IP-10) between male HIV-infected cannabis users (HIV + MJ+) and non-cannabis users (HIV + MJ−) and determine whether in vitro THC affected CD16 expression as well as IP-10 production by monocytes. Cannabis use was determined by self- reporting and confirmed by serum detection of THC metabolites using a THC ELISA (RTU) Forensic Kit.	Patients smoked their own cannabis	HIV + MJ+ donors had lower levels of circulating serum IP-10 and CD16+ monocytes compared to HIV + MJ − donors, suggesting anti-inflammatory effects due to cannabis consumption.	N/A	[244]

### 2.4. Neuroinflammation

The reciprocal interactions between the CNS and the immune system are well known. Indeed, through an array of chemical messengers (i.e., neurotransmitters, neurotrophic factors, and eCBs), the CNS is able to regulate the functions of the various immune cells, such as astrocytes and microglia [245,246]. Conversely, these neuronal bioactive molecules can be regulated by cytokines and other small molecules, such as nitric oxide, which are the main effectors of immune systems [247]. These bidirectional interactions are fundamental for appropriate adaptive cellular responses, such as inflammatory processes, which play significant roles in pathological states. Besides neurodegenerative diseases, a growing body of evidence has demonstrated a strong contribution of neuroinflammation to the onset and severity of several neuropsychiatric diseases, including drug addiction [246,248].

The eCBS, modulating both synaptic transmission and neuroinflammation, has been recently proposed as a key modulator of neuroinflammation [249,250]. Consistently, accumulating preclinical evidence has revealed the immune-modulatory effects of cannabinoids, either anti- or pro-inflammatory, based on the specific molecules, dosage, duration, and age of exposure. For instance, THC in combination with CBD showed neuroprotective effects in various preclinical models of neuroinflammation, such as multiple sclerosis [251] and Alzheimer’s disease [252], and in brain injuries, such as ischemia (see review [253]); chronic high-dose THC or synthetic cannabinoid exposure induced neurotoxic and neuroinflammatory effects in different animal models [254,255,256,257,258,259].

Importantly, few studies have reported that chronic THC exposure during adolescence induces neuroinflammatory phenotypes in rodents. Indeed, Zamberletti and colleagues (2015) reported that the cognitive impairments and behavioral effects induced by adolescent Δ9-THC exposure in female rats were associated with long-term neuroinflammatory effects characterized by altered microglial morphology; increased expression of pro-inflammatory cytokines, such as TNF-α, inducible Nitric Oxide Synthase (iNOS), and Cyclooxygenase-2 (COX-2); and a reduction in the anti-inflammatory cytokine IL-10 [260]. Interestingly, THC-induced microglial activation was region-specific, since no alterations were detected in the nucleus accumbens, hippocampus, or amygdala. The same authors reported sex differences in the brain regions affected and the profiles of pro-neuroinflammatory biomarkers induced by THC adolescent exposure. Indeed, an alteration in astrocyte reactivity (i.e., increased GFAP levels) has been reported in the hippocampi of male rats after adolescent Δ9-THC treatment. Moreover, astrocyte activation was associated with increased protein expression of TNF-α and iNOS, together with a concomitant reduction in the anti-inflammatory cytokine IL-10 [261].

Although only a few preclinical studies have evaluated this aspect, THC exposure during adolescence seems to promote neuroinflammation in adulthood. Indeed, Moretti and colleagues (2014) observed in mice a decrease in the proinflammatory cytokines IL-1β and TNF-α and an increase in the anti-inflammatory cytokine IL-10 produced by macrophages in both adolescent and adult mice 24 h after THC chronic treatment (5–15 mg/kg s.c., 10 days). Interestingly, the mice treated during adolescence displayed in adulthood a proinflammatory macrophage phenotype (IL-1β and TNF-α were elevated; IL-10 was decreased) with blunted production of Th cytokines, suggesting that THC in adolescent mice triggers immune dysfunctions that last long after THC consumption, switching the immune system to a proinflammatory status in adulthood [262]. The same authors subsequently demonstrated a similar effect on brain cytokines [263], pointing out that THC exposure during adolescence leads to vulnerability to immune and behavioral diseases in adulthood.

On the other hand, accumulating evidence demonstrated the anti-inflammatory action of CBD in multiple neuroinflammatory animal models. For instance, CBD exhibited neuroprotective effects in neurodegenerative diseases, such as Alzheimer’s disease [5,264]. Recently, it has been demonstrated that chronic CBD (50 mg/kg) tended to reduce insoluble Aβ_40_ levels in the hippocampi of transgenic mice modeling Alzheimer’s disease, even though no effects on neuroinflammation, neurodegeneration, or PPARγ markers in the cortex were observed [265]. Furthermore, another study showed that CBD treatment in a mouse model of Alzheimer’s disease was able to successfully attenuate neuroinflammation, simultaneously improving mitochondrial function and ATP production via TRPV2 activation [266].

CBD exhibits neuroprotective effects also in Parkinson’s disease. For instance, chronic treatment with CBD (10 mg/kg, i.p., 28 days) reduces nigrostriatal degeneration and neuroinflammatory response and improves motor performance in a Parkinson’s disease rat model. Notably, CBD exhibits a preferential action on astrocytes, enhancing the endogenous neuroprotective response of ciliary neurotrophic factor through its activity on TRPV1 receptors [267].

The anti-inflammatory efficacy of CBD treatment has also been reported for other neurodegenerative diseases, such as epilepsy [268,269] and multiple sclerosis [270,271,272]. Similarly, CBD shows antioxidant and neuroinflammatory effects in cases of ischemic insult, reducing glutamate excitotoxicity, oxidative stress, and glial response, both in neonatal and adult rodents [240,269,273,274,275,276,277,278,279,280].

The preclinical efficacy of CBD in neuroinflammatory models has been only partially reported at clinical levels, in combination with THC (see reviews [281,282,283]). As a matter of fact, a medical formulation containing equivalent concentrations of CBD and THC (50:50) called Sativex^®^ (GW Pharmaceuticals, UK) is a cannabis-based therapeutic approved for the treatment of pain and spasticity in multiple sclerosis [284].

However, while a number of preclinical studies have investigated the effects of THC and CBD on the neuroinflammatory components of multiple pathologies, few clinical studies have looked into these aspects. In this regard, it has been reported that HIV-infected cannabis users showed beneficial reductions in systemic inflammation and immune activation in the context of antiretroviral-treated HIV infection (see Table 2) [243,244].

Despite the therapeutic efficacy, the specific mechanisms underlying the anti-inflammatory/neuroprotective effects of CBD and CBD-THC products are not yet fully understood and most likely involve receptor-independent mechanisms, such as nuclear factors (see review [285]). To date, no in vivo evidence exists about anti-inflammatory CBD effects in neuropsychiatric disorders, such as schizophrenia, or in wild-type (animals) or healthy conditions (humans). Similarly, there is a lack in the literature of reports on the neuroprotective or neuroinflammatory effects of CBD and CBD-THC combination exposure in adolescence. A few preclinical studies have reported modulatory action of THC and CBD on neuroinflammation caused by alcohol and other substance abuse exposure. For instance, it has been suggested that CBD and THC can be effective against methamphetamine-induced mitochondrial dysfunction, neuroinflammation, and neurodegeneration, in both human and animal subjects via the TLR4/NF-kB signaling pathway [286]. Moreover, CBD is able to restore the increased levels of TNFα and IL-6 in the hippocampi of mice early exposed to alcohol [287].

Although preclinical studies suggest the potential effectiveness of CBD in neuroinflammation, clinical evidence is still lacking, and further investigations are needed. Therefore, a deep characterization of the impact of CBD and other cannabinoids on neuroinflammatory responses may represent an important strategy to develop new pharmacological tools to treat multiple neuroinflammatory pathologies.

## 3. Materials and Methods

A systematic literature review was conducted and documented in accordance with the Preferred Reporting Items for Systematic Reviews and Meta-Analyses (PRISMA) guidelines [288]. PubMed was the main database consulted and was searched by combining the following words: (THC (Title/Abstract)) and (adolescence (Title/Abstract)) OR (CBD (Title/Abstract)) and (adolescence (Title/Abstract)), (cannabis exposure (Title/Abstract)) and (adolescence (Title/Abstract)) OR (clinical/preclinical trial (Title/Abstract)) and (adolescence (Title/Abstract)). Restrictions on the year of publication were applied, as only papers published between 2015 and 2022 were considered.

## 4. Conclusions

In conclusion, all the data collected in this review suggest that there is a huge gap in the literature on the rewarding and addictive effects of CBD and on how its consumption in adolescence might affect the development of SUDs. Despite some findings obtained in adults suggesting anxiolytic and antipsychotic effects of CBD, these effects have not been tested in adolescents yet. Similarly, there is a lack of information in the literature about the neuroprotective or neuroinflammatory effects of CBD and THC-CBD combination exposure in adolescence, especially in terms of clinical research. Notably, some clinical studies focus more on how THC and CBD affect the symptoms of multiple pathologies but they do not investigate the effects on biological markers typical of these diseases. Additionally, the effects of CBD on brain cognition are controversial, especially in preclinical studies. These discrepancies are due to the CBD doses, ratios of THC:CBD, routes of administration, models (humans, primates, or rodents), and tasks used. A big limitation is the variability of the cannabis products used in clinical trials; having worldwide guidelines for cannabis clinical trials may represent a key factor in obtaining more interpretable results. Additionally, while the randomization of participants is important for the comparison of results, it does not allow for the representation of multiple aspects of normal life that could influence the outcomes of using substances such as THC and CBD. Therefore, a better preclinical and clinical characterization of CBD is necessary to deeply understand the impact of CBD treatment and its potential therapeutic effects, especially in adolescence. Notably, unraveling the preclinical and clinical effects of both THC and CBD action on multiple non-cannabinoid receptors may represent a key factor in developing new pharmacological tools to treat some pathologies.

## Figures and Tables

**Figure 1 ijms-24-05251-f001:**
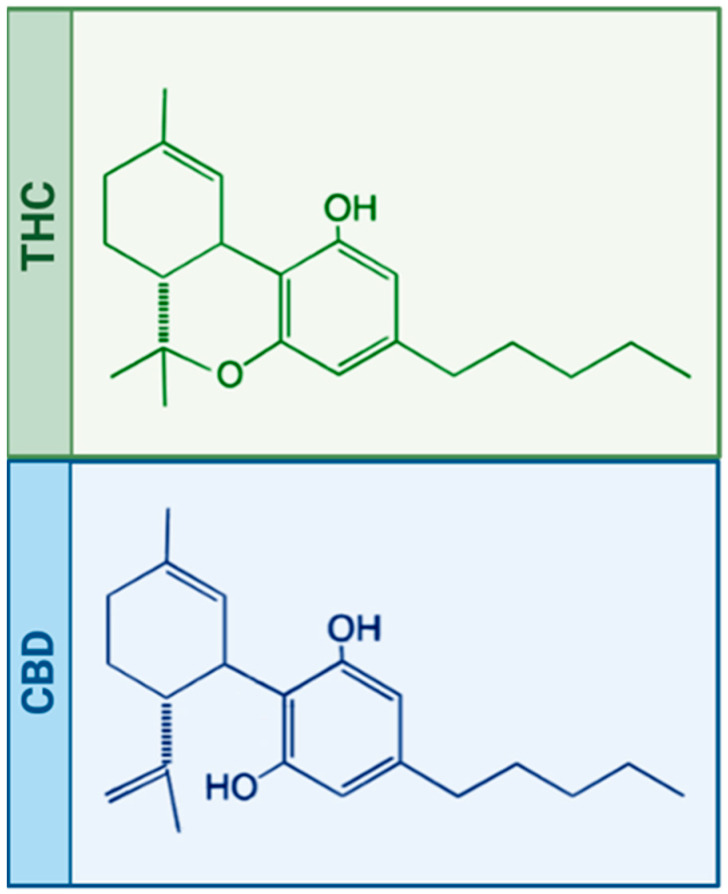
Chemical structures of Δ9-tetrahydrocannabinol (THC) (green panel) and cannabidiol (CBD) (blue panel).

**Figure 2 ijms-24-05251-f002:**
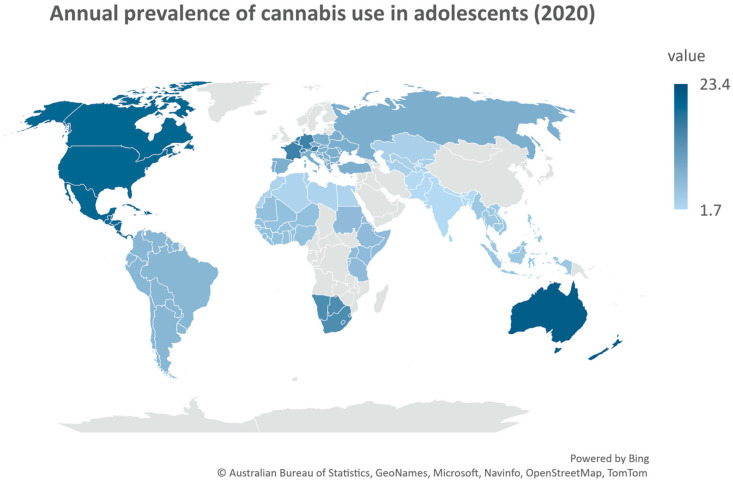
A blue-scale representation of the prevalence (percentage) of cannabis use in adolescents (aged 15–16) by country in 2020. Data were adapted from World Drug Report 2022, Statistical Annex Tables.

**Figure 3 ijms-24-05251-f003:**
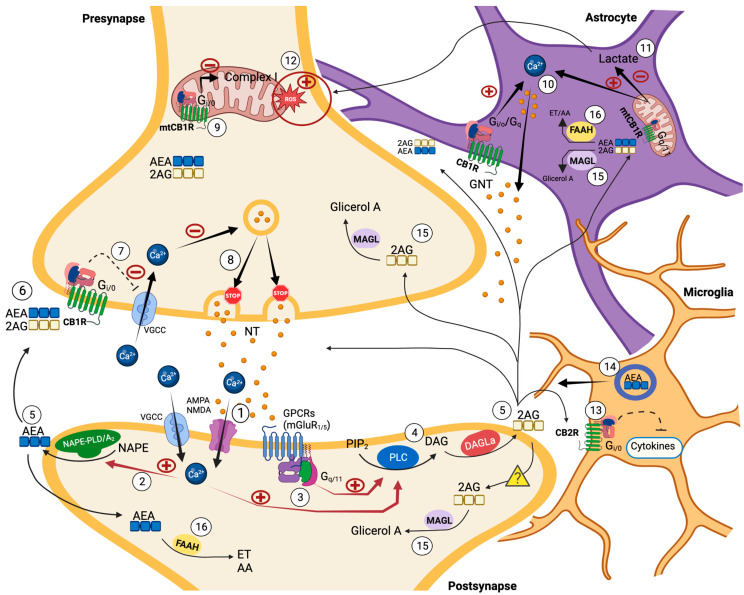
Model of endocannabinoid (eCBs) signaling at central synapses. In neurons: (1) The activation of ionotropic receptors (e.g., α-amino-3-hydroxy-5-methyl-4-isoxazolepropionic acid receptor (AMPA) and N-methyl-D-aspartate receptor (NMDA)) on postsynaptic neurons by neurotransmitters (NTs) and/or the depolarization of the neuronal membrane triggers the entry of Ca^2+^ through the voltage-gated Ca^2+^ channels (VGCCs). (2) The increase in intracellular Ca^2+^ activates the enzymes involved in the synthesis of arachidonoyl ethanolamide (AEA), including NAPE-phospholipase D (NAPE-PLD) and/or phospholipase A2. (3) The entry of Ca^2+^ through the aforementioned mechanisms and/or the activation of G protein-coupled receptors (GPCRs) coupled to Gq protein alpha subunits (Gαq/11s) (e.g., group I metabotropic glutamate receptors (mGluRs) and muscarinic acetylcholine receptor (mAChR) subtypes M1/3) by neurotransmitters (NTs) activates phospholipase C (PLC), (4) which hydrolyzes phosphatidylinositol 4,5-bisphosphate (PIP2) into diacylglycerol, which is converted into 2-Arachidonoylglycerol (2-AG) by diacylglycerol lipase α (DAGLα). (5) Following their de novo synthesis, AEA and 2-AG are released into the synaptic space. (6) Once released, they travel “backwards” and activate the cannabinoid receptor 1 (CB1R) Gα_i/o_-GPCRs located on presynaptic neurons, (7) which through G_i/0_ proteins block the VGCCs, resulting in (8) inhibition of the release of NTs by the presynaptic neuron. (9) In the presynaptic neuron, AEA and 2-AG also activate mitochondrial CB1Rs (mtCB1Rs) with probable consequent inhibition of complex I activity [58], which modulates the organelle’s respiration and energy production. In glial cells, on astrocytes the presence of CB1R has been reported on both the cell membrane (Ref. 41F) and the mitochondrial membrane [59]. (10) The activation of CB1R and mtCB1R leads to an increase in Ca^2+^, which promotes the release of gliotransmitters (GNTs) into the synapse. Furthermore, (11) the activation of mtCB1R leads to a reduction in the glycolytic production of lactate in astrocytes, which results in (12) redox stress in the presynaptic neuron [60]. (13) On microglia, eCBs activate cannabinoid receptors 2 (CB2Rs) by blocking the production of cytokines [61]. (14) Moreover, AEA released from microglia via macrovesicles may contribute to endocannabinoid signaling at the central synapse [62]. eCB catabolism: (15) 2-AG and (16) AEA are inactivated by monoacylglycerol lipase (MAGL) and fatty acid amide hydrolase (FAAH), respectively, in glycerol A and ethanolamine + AA in both astrocytes and neurons [62,63,64].

**Figure 4 ijms-24-05251-f004:**
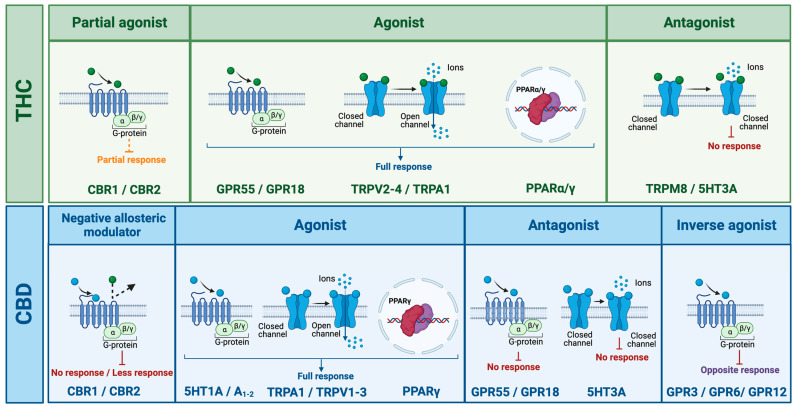
Δ9-tetrahydrocannabinol (THC) (green) and cannabidiol (CBD) (blue) activity on cannabinoid and non-cannabinoid receptors. Abbreviations: Cannabinoid receptors 1 and 2 (CB1R and CB2R); peroxisome proliferator-activated alfa and gamma (PPAR α-γ) receptor; orphan G-protein coupled receptors 3, 6, 12, 18, and 55 (GPR3, GPR6, GPR12, GPR18, and GPR55); transient receptor potential of vanilloid 1-4 (TRPV1-4) and ankyrin (TRPA1) channels; TRP cation channel subfamily M member 8 (TRPM8); serotonin 1A and 3A receptor (5-HT3A and 5-HT1A); adenosine receptors 1 and 2 (A_1_ and A_2_).

## Data Availability

Not applicable.

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
