# Peer review of "THC and CBD: Villain versus Hero? Insights into Adolescent Exposure"

_ijms, 2023, doi:10.3390/ijms24065251_

Round 1
Reviewer 1 Report
Review of a manuscript “THC and CBD: Villain vs Hero? Insights into adolescent exposure” by Nicholas Pintori and coauthors submitted to IJMS.
Cannabis may be used as a medicinal herbal medication to treat various symptoms and as a recreational drug. However, the results of the research show that chronic cannabis use is harmful to the central nervous system (CNS), respiratory system, and cardiovascular system. The harm is exceptionally high if Cannabis. if used in adolescence. The authors collect preclinical and clinical evidence about the effects of cannabidiol on human health. This is an important biomedical area, and the review would be interesting for the readers of the IJMS. The following corrections and additions should be made.
Abstract
“These two compounds are remarkably similar yet vastly different in affecting the brain.” The authors should be more specific in writing about the similarity between the two compounds. For example, they may write:” These two compounds have remarkably similar chemical structures”.
Figure 1
The size of the figure 1 may be reduced.
1.2. Epidemiology: use in adolescence. The title should be rewritten as “1.2. Epidemiology: use of cannabis in adolescence”.
2.1 “These opposite effects are due to the different pharmacological properties showed by these two cannabinoids.” Should be corrected as “These opposite effects are due to the different pharmacological properties shown by these two cannabinoids.”
2.3. Neuropsychiatric disorders.
The authors should begin this section by adding the following introductory sentence and a reference: ”A wide range of phytochemicals possess biological activity affecting a variety of neurological and neuropsychiatric disorders” [reference “Phytochemicals as regulators of genes involved in synucleinopathies”. Biomolecules, 2021, 11 (5), 624.
“which may play an important role on this CBD’s action.” Should be corrected as :” which may play an important role in this CBD’s action”
These two sentences sounds as contradictory: 1. ”Starting from the 90s growing clinical evidence confirms the strong association between adolescent cannabis use and schizophrenia, or other psychotic disorders later in life (see for review [162]).”
2. “Noteworthy, some findings suggest no causal link between early cannabis consumption and likelihood of neuropsychiatric disorders in adulthood [164, 165].”.
The authors should express their sense in a less contradictory way.
“The preclinical efficacy of CBD in neuroinflammatory models, has also been reported at the clinical levels, in combination with THC.: A reference should be added here.
Overall: interesting and helpful review.
Reviewer 2 Report
Despite binding the same CNS receptors, Δ9-tetrahydrocannabinol (THC) is psychoactive, while cannabidiol (CBD) has anxiolytic and antipsychotic properties. In the current review, the authors summarize the preclinical and clinical evidence about the effects of CBD. They list prior studies on the epidemiology of cannabis use, THC and CBD pharmacology, and the divergent central effects of THC and CBD (including reward and substance use disorders (SUD), neuropsychiatric disorders, memory and attention, and neuroinflammation). The authors provided four interesting figures that summarize the chemical structures of THC and CBD, prevalence of cannabis use in adolescents, model of endocannabinoid signaling at central synapses, and activity of THC and CBD on cannabinoid and non-cannabinoid receptors. From this discussion, the authors suggest that there is a huge gap in the literature on the CBD’s rewarding and addictive effects, and how its consumption in adolescence might affect the development of SUDs. While the summarized information is interesting and the review is well-written, the authors do not provide take-home messages at the end of the respective section for the reader.
Comments:
1) The provided sections read like narration for the evidence of discussed points without critical aspects/reflection points. At the end of each section, a take-home message is advised to be provided.
2) To avoid readers’ confusion, the authors are advised to clearly describe in the narration of previous literature whether these data are derived from clinical studies or from experimental studies. This point needs to be carefully addressed by the authors in the entire manuscript.
3) The implications of THC and CBD in different disorders should be summarized in different tables with adequate details about study authors, the used doses, sample/design, findings/primary outcomes, adverse events (if any), and retention in trial.
4) The work lacks future directions that include limitations and what is the next step to translate these findings to clinical settings.
5) The authors are advised to make the figure captions stand-alone. To this end, authors are advised to provide the full names of all the listed abbreviations in the figures. For example, in figure 1, the full names of THC and CBD need to be described.
6) The image in figure 2 needs to be properly adjusted/resized. As it currently stands, there is no provided explanation of the different colors displayed in the image.
7) More recent 2022 references are advised to be added to the review.
8) Careful revision of the reference list should be performed. For example, reference no. 5 lacks the year, title, and other essential data.
9) Reference no. 9 lacks the page range.
10) The authors are advised to unify the way they write the journal name. Sometimes it is written as an abbreviation (such as reference no. 1) while in other references it was written as a full name (such as reference no. 9). Please, follow the journal instructions in this regard.
11) The authors are advised to correct the typo in the title of manuscript that reads as “THC and CBD: Villain vs Hero? Insights into adolescent exposure”. The authors may consider correcting it by either using “versus” or “vs.”.
12) there is a gap in numbering section 2 (Divergent central effects of THC and CBD), section 2.1. is directly followed by section 2.3. Please, correct this issue.
13) In the introduction section, the authors state “… but it was only in the 800 A.D. that smoking cannabis became more common in the Middle East and South Asia, mainly due to the spread of Islam”. The authors are advised to remove the above statement as it calls for hate of religions and it adds no real value to the scientific evidence provided in the present review.
